# Visible Light Photoelectrochemical Sensor for Dopamine: Determination Using Iron Vanadate Modified Electrode

**DOI:** 10.3390/molecules27196410

**Published:** 2022-09-28

**Authors:** Luan Pereira Camargo, Marcelo Rodrigues da Silva Pelissari, Paulo Rogério Catarini da Silva, Augusto Batagin-Neto, Roberta Antigo Medeiros, Marcos Antônio Dias, Luiz Henrique Dall’Antonia

**Affiliations:** 1Departamento de Química/CCE/UEL, State University of Londrina (UEL), Londrina 86057-970, Brazil; 2Graduated Program in Chemistry, State University of Londrina (UEL), Londrina 86057-970, Brazil; 3National Institute of Science & Technology in Bioanalytic (INCTBio), Campinas 13083-970, Brazil; 4Faculdade de Engenharia, CTI, São Paulo State University (UNESP), Bauru 17033-360, Brazil; 5Departamento de Física/CCE/UEL, State University of Londrina (UEL), Londrina 86057-970, Brazil; 6Instituto de Ciências e Engenharia, São Paulo State University (UNESP), Itapeva 18409-010, Brazil; 7Departamento de Neurocirurgia, Centro de Ciências da Saúde, State University of Londrina (UEL), Londrina 86057-970, Brazil

**Keywords:** levodopa, non-enzymatic sensor, ITO electrode, FeVO_4_, Fe_2_V_4_O_13_

## Abstract

This study reports a facile approach for constructing low-cost and remarkable electroactivity iron vanadate (Fe-V-O) semiconductor material to be used as a photoelectrochemical sensor for dopamine detection. The structure and morphology of the iron vanadate obtained by the Successive Ionic Adsorption and Reaction process were critically characterized, and the photoelectrochemical characterization showed a high photoelectroactivity of the photoanode in visible light irradiation. Under best conditions, dopamine was detected by chronoamperometry at +0.35 V vs. Ag/AgCl, achieving two linear response ranges (between 1.21 and 30.32 μmol L^−1^, and between 30.32 and 72.77 μmol L^−1^). The limits of detection and quantification were 0.34 and 1.12 μmol L^−1^, respectively. Besides, the accuracy of the proposed electrode was assessed by determining dopamine in artificial cerebrospinal fluid, obtaining recovery values ranging from 98.7 to 102.4%. The selectivity was also evaluated by dopamine detection against several interferent species, demonstrating good precision and promising application for the proposed method. Furthermore, DFT-based electronic structure calculations were also conducted to help the interpretation. The dominant dopamine species were determined according to the experimental conditions, and their interaction with the iron vanadate photoanode was proposed. The improved light-induced DOP detection was likewise evaluated regarding the charge transfer process.

## 1. Introduction

Dopamine (3,4-dihydroxyphenethylamine, DOP) is one of the most know brain neurotransmitters [1,2]. It belongs to the catecholamine and phenethylamine groups and acts in the human body’s cardiovascular, renal, and hormonal systems, in addition to as a messenger in the nerve cells to communicate with each other [1,2,3,4,5]. Abnormal concentration levels can be associated with several disorders. The DOP biomolecule is vital to various neuronal functions in the human body, such as memory, learning, attention, perception, emotion, cognition behavior, and movement control [1,2,3,4]. Consequently, DOP’s low concentration levels may be responsible for serious diseases, such as attention deficit hyperactivity disorder (ADHD), epilepsy, schizophrenia, Tourette’s syndrome, and Parkinson’s disease, among other [5,6]. At the same time, a high level may be correlated with systemic arterial hypertension, cardiotoxicities, heart failure, and drug addiction [5,6]. Generally, the treatment of Parkinson’s disease includes Prolopa^®^ BD (levodopa and benserazide) capsules administration [7]. Levodopa, also known as *L*-dopa or L-(-3-(3,4-dihydroxy-phenyl)-alanine, has a very similar structural formula to dopamine; the difference is a carboxylic group on the dopamine structure (please consult Appendix A). *L*-dopa is an amino acid precursor of dopamine and, in many tissues, can be decarboxylated to the active moiety [7,8]. At the same time, the benserazide (Appendix A) is an inhibitor of aromatic amino acid decarboxylase, diminishing the degradation of levodopa [9]. Therefore, levodopa is largely available in the human body, especially in the brain capillaries of the brain where it is converted into dopamine [9,10]. Furthermore, the levodopa and benserazide combination reduces the side effects of levodopa on the human body, such as nausea, vomiting, confusion, hallucinations, and possibly cardiac arrhythmias [9].

Conventional methods used to detect and quantify dopamine levels include two-step processes. Firstly, DOP concentration is quantified using either an enzyme-linked immunosorbent assay (ELISA) or high-performance liquid chromatography [11,12]. Then, fluorometry and mass spectroscopy techniques are used for the detection step [11,12]. Nevertheless, these methods require a large number of reagents, specific apparatus, operation abilities, long testing times, waste-containing organic solvents being produced, and expensive costs [11,13]. Therefore, alternative procedures are developed by researchers to overcome these limitations. The most popular procedures include electrochemical and biological sensors (biosensors) [14]. The biosensors are characterized by a biological element (enzymes, antibodies, aptamers, or nucleic acids) for target recognition and a transducer responsible for converting the biological response into an electrical signal [11,14]. Although these devices have high catalytic efficiency, excellent specificity, and sensibility, and the analyte can be detected faster than conventional methods, some drawbacks must be considered. In general, the biomolecules approach may have a high cost, low thermal and chemical stability, in addition to a complex immobilization procedure [14,15,16].

Electrochemical detection using surface-modified electrodes with metallic and oxide materials is frequently used to overcome the limitations of biosensors. The direct advantages include low-cost configuration equipment, rapid detection, easy electrode construction, and good stability and reproducibility. In the study developed by Anshori et al. [17], an exciting approach was used to detect dopamine levels in phosphate buffer saline (PBS) solution. Nanoparticles of graphene oxide were combined with Fe_3_O_4_, obtaining a nanocomposite with a remarkable ability to detect dopamine. Specifically, DOP was detected in the range of 1–10 μmol L^−1^ with a limit of detection (LOD) and quantification (LOQ) of 0.48 μmol L^−1^ and 1.6 μmol L^−1^, respectively. Trindade et al. [18] reported a novel electrochemical methodology for DOP detection in synthetic human urine. Renewable carbon from bamboo biomass was modified with copper nanoparticles and used as a modifier of the glassy carbon electrode (GCE). The linear range obtained for DOP was 0.05–5.0 μmol L^−1^ with a LOD of 0.04 μmol L^−1^. To improve the performance during the DOP detection (still using the GCE), a combination of a metal oxide with a covalent organic framework was proposed by Chu et al. [19]. The authors prepared a core-shell structured composite by encapsulating CuO into the TAPB-DMTP-COF host matrix. As a result, an ultrasensitive electrochemical sensor was formed in PBS solution, with a linear range between 0.07 and 800 μmol L^−1^, and a LOD of 0.023 μmol L^−1^. Other manuscripts also stand out in the electrochemical detection of DOP [20,21,22].

Besides, using a specific light source with semiconductors materials play a vital role in enhancing the electrochemical signal [23,24]. Compared with traditional electrochemistry, photoelectrochemical (PEC) detection (photo-assisted electrochemical sensors) can achieve a combination of advantages with photocatalysis and electrocatalysis, such as an ultra-sensitivity and a lower background signal [25]. When the system is irradiated by a light source with energy greater than the band gap energy (*E*_g_) of the semiconductor, electrons from the valence band (VB) are promoted to the conduction band (CV), giving rise to the photogenerated electron/hole pair (*e*^−^_CB_/*h*^+^_VB_) [15,23,25,26,27]. The photogenerated holes (*h*^+^_VB_) on the photoelectrode surface will give rise to hydroxyl radicals (^•^OH) from the reaction with the water [28,29]. At the same time, the applied potential minimizes the natural recombination of the *e*^−^_CB_/*h*^+^_VB_ in semiconductors [28,29,30]. In this sense, in a PEC system, dopamine detection can occur from the analyte oxidation with the hydroxyl radicals and by the electrochemical oxidation (electrode surface) [31,32]. The response is measured by the photocurrent generated during applying potential and light irradiation. On the other hand, Wang et al. studied a novel strategy for the construction of PEC sensors [33]. The authors showed possible dopamine detection based on photoinduced electron transfer between an electron acceptor (in this case, benzoquinone was used) and a semi-conductor material (CdS quantum dots) [33]. The strategy could detect dopamine with high selectivity and sensitivity [33].

Several types of semiconductors and oxide-based heterostructures emerge as electroactive materials to be used as an electrode in PEC cells, such as BiVO_4_/FeOOH [24], CdS nanoparticles [25], BiVO_4_/GQDs [31], ferroelectric perovskite oxide@TiO_2_ [32], BiVO_4_ [34], Au@WP5/BiOBr [35], CdSe/TiO_2_ [36], graphene quantum dots + TiO_2_ [37], nanoMoS_2_ modified gold electrode [38], and many others [39,40]. However, some essential criteria should be adopted to maximize the performance of PEC sensors. The fundamental factors are the electrode material characteristics, chemical stability in aqueous solution, resistance to electrochemical and photoelectrochemical corrosion, non-toxicity, energy absorption capacity, low-cost synthesis process, and good absorption of visible light. Thus, iron vanadate semiconductor materials (FeVO_4_ and Fe_2_V_4_O_13_) have attracted substantial interest in this context. Low band-gap energy (visible light absorption), excellent chemical and thermal stability, low toxicity, and the elemental abundance of iron and vanadium in the earth’s crust make it an attractive and very low-cost material to be used in PEC cells [15,41,42,43].

Therefore, this work aims to evaluate the electrocatalytic and photoelectrocatalytic activity of the non-enzymatic iron vanadate electrode. The Successive Ionic Adsorption and Reaction (SILAR) process obtained the material with different amounts of layers (5, 10, and 15), and the dopamine was used as a probe for the catalytic activity. This proposal presents an exciting approach for high-performance PECs with low-cost materials and high selectivity, sensitivity, and stability. The physicochemical characterization was performed through X-ray diffraction (XRD), Fourier-transform infrared spectroscopy (FTIR), diffuse reflectance, and scanning electron microscopy (SEM) techniques, using the solid and film form sample (five layers). Besides, electrochemical measurements in the absence and presence of visible light were also performed (electrodes with 5, 10, and 15 layers were evaluated). Cyclic voltammetry (CV), differential pulse voltammetry (DPV), chronoamperometry, and electrochemical impedance spectroscopy (EIS) techniques were used. DFT-based electronic structure calculations were also conducted to help the interpretation. To the best of the authors’ knowledge, the iron vanadate used as photoanode material for the photoelectrochemical oxidation of dopamine has not yet been reported in the literature. Moreover, a systematic study of the electroanalytical parameters (limit of detection (LOD), limit of quantification (LOQ), sensibility, and stability) was likewise developed. The accuracy of the proposed electrode was assessed by determining dopamine in the presence of organic compounds and using artificial cerebrospinal fluid as the electrolyte solution. Furthermore, Prolopa^®^ BD was chosen as a commercial drug because of the structural similarities between levodopa and dopamine.

## 2. Results and Discussion

### 2.1. Structural and Morphological Characterization

The characterization of molecular and morphological structure and identification of the sample obtained by the SILAR process, Fourier-transform infrared spectroscopy (FTIR), X-ray diffraction (XRD), and scanning electron microscopy techniques were used. Furthermore, the band gap energy of the samples was estimated by UV–Vis reflectance diffuse. These characterizations were performed with the powder and film iron vanadate (five layers deposited on the ITO conductive substrate).

Infrared spectra were obtained for the powder and film samples, as shown in Figure 1a. The region highlighted in the graph and amplified, covering the wavenumber range between 500 and 1050 cm^−1^, shows the typical bands of the FeVO system. The bands are assigned to four regions (I, II, III, and IV) (Figure 1a). In region I, from 880 to 1050 cm^−1^, there is the terminal stretching of the V–O bond. At the same time, region II (from 700 to 880 cm^−1^) can be associated with the bridge-type stretching of the V–O· · ·Fe system. Region III encompasses stretches of the V–O· · ·Fe and V· · ·O· · ·Fe systems. Finally, below 550 cm^−1^, there is the deformation mode of the V–O–V system and stretching of the Fe–O bond [44,45,46,47,48]. There is no significant difference between the powder and film FTIR spectra (Figure 1a). Besides, the powder XRD result confirmed the formation of the iron vanadate material (diffractogram given in the Appendix A). All the diffraction peaks obtained were associated with FeVO_4_ triclinic structure (PDF 01-071-1592) and Fe_2_V_4_O_13_ monoclinic structure (PDF 01-089-5460). The strongest triclinic structure peaks occurred at 10.1°, 13.7°, 16.6°, 20.3°, 25.0°, 27.1°, and 27.7°, which were associated with the [0 0 1], [0 −1 1], [0 1 1], [1 −1 1], [0 1 2], [−2 0 1], and [1 −1 2] crystallographic planes. At the same time, the monoclinic structure shows the strongest peaks at 12.4°, 17.4°, 21.7°, 22.7°, 23.8°, and 26.1°. These correspond to the [0 0 2], [1 1 −2], [1 1 −3], [0 2 2], [2 1 0], and [0 1 4] crystal planes, respectively. Furthermore, there are no peaks regarding iron oxide structures, such as Fe_2_O_3_ and Fe_3_O_4_.

In the sequence, to evaluate the optical properties and estimate the band gap energy (*E*_g_), diffuse reflectance spectroscopy was used. Figure 1b shows the UV–Vis spectrum of the powder and film samples recorded in the wavelength range between 400 and 900 nm. According to the reflectance profile, Figure 1b, the material presents optical absorption in the visible region of the electromagnetic spectrum. The estimated *E*_g_ value was obtained from the Wood–Tauc model, according to Equations (1) and (2) [49,50]:(1)$$\alpha =F\left(R\right)=\frac{{\left(1-R\right)}^{2}}{2R}$$
(2)$$\left(\alpha h\upsilon \right)=A{\left(h\upsilon -E_{g}\right)}^{1/n}$$
where *F*(*R*) represents the absorption coefficient, *α*, obtained by the Kubelka–Munk function (used to transform reflectance spectra to the corresponding absorption spectra [50]), *R* is the absolute reflectance, *h* is the Planck constant, *ν* is the frequency of light, *hν* represents the photon energy, *A* is a constant, *E*_g_ is the band gap energy, and *n* is related to the nature of the electronic transition that occurs in the semiconductor. In this work, *n* was considered a direct allowed transition, equal to 1/2 (Equation (2)) [42,49,51,52]. The band gap energy was estimated from the extrapolation of the linear decreasing region of the (*αhν*) as a function of *E*_g_ (inset Figure 1b), taking into account the absorption baseline, as mentioned by Makuła et al. [49]. For the powder and film samples, 2.28 and 2.30 eV were estimated, respectively. Thus, there is no significant difference between the values (Figure 1b), which was very similar to some previously published manuscripts [15,41,42,52].

SEM images of the sample in the film form (five layers) can be seen in Figure 2. Surface images at different degrees of magnification, 5000×, 12,000×, and 30,000×, reveal a surface homogeneity in the distribution of spherical-shaped particles. A few agglomerates of particles can also be observed. The result is in good agreement with other manuscripts [15]. A cross-sectional image, with a magnification of 50,000×, reveals a good dispersion and homogeneity in the deposited film, with a thickness of approximately 280 nm (Figure 2).

### 2.2. Electrochemical and Photoelectrochemical Characterization

After the structural and morphological characterizations, the iron vanadate electrodes had their electrochemical and photoelectrochemical properties evaluated. Cyclic voltammetry (CV), chronoamperometry, Mott–Schottky, and electrochemical impedance spectroscopy (EIS) techniques were used. From here, the electrodes with 5, 10, and 15 layers were identified as ITO/FeVO(I), ITO/FeVO(II), and ITO/FeVO(III), respectively. A visible light source (35 W Xe lamp–Vision HIDlamp) was used when necessary. All current values (*j*) were normalized by the electroactive area (*j*_N_), according to the Randles–Sevcik equation [53] (for more details, please consult the Appendix A and Table 1).

Initially, the cyclic voltammograms revealed the prepared film’s photoelectroactivity. The results obtained by CV curves (Figure 3a) showed the electroactivity for the ITO/FeVO(I) electrode in the dark condition (solid black line), under continuous visible light illumination (solid red line), and visible light–chopped illumination (10 s on/off, solid blue line). For the lower potential range (between 0 and ~0.20 V), the current density (*j*_N_) responses are negligible compared to those obtained under photoelectrochemical conditions, and there is no distinguished difference between dark and visible light conditions (*j*_N_ less than 1.0 µA cm^−2^). As the anodic scan progressed, an augmentation in the photoinduced current was observed. This behavior can be associated with the electronic excitation process of a semiconductor. During light incidence, the electrons overcome the band gap energy (*E*_gap_) and are promoted to an excited energy level, resulting in increased current and the formation of electron/hole pair (*e*^−^_CB_/*h*^+^_VB_) [15,23,25,26,27]. Furthermore, in the anodic potential scan under transient condition (chopped illumination), when the light is turned on, current density reaches almost the same value as the curve obtained with continuous light (Figure 3a). At the same time, when the light is turned off, *j*_N_ drops to the same value as the dark condition curve (Figure 3a), indicating an *n*-type semiconductor behavior. The substrate response under visible light illumination was also plotted (Figure 3a—dot purple line). However, the *j*_N_ can be negligible in the potential range (*j*_N_ less than 1.0 µA cm^−2^). The same profile was observed for the ITO/FeVO(II) and ITO/FeVO(III) electrodes, as shown in Appendix A.

The chronoamperometry technique was used in the sequence to compare the electrodes’ photocurrent stability. First, the visible light was employed in a transient condition under the application of different potentials (0, 0.2, 0.4, 0.6, 0.8, and +1.0 V (vs. Ag/AgCl)) in the dark for 50 s, during light incidence (also for 50 s) and another 50 s in the dark condition (Appendix A). Sodium sulfate (0.1 mol L^−1^) was used as the electrolyte solution. From these measurements, it was possible to observe a significant increase in photocurrent density for all electrodes, especially from the potential of +0.6 V. An increase of 14, 8, and 2 times was verified in the *j*_N_ when compared to the potential of +0.4 V for the electrodes ITO/FeVO(I),-(II),-(III), respectively (Appendix A). This behavior was similar to that obtained in the cyclic voltammograms, confirming the *n*-type semiconductor behavior.

Besides, due to the highest *j*_N_ values, the limit potential of +1.0 V was selected to evaluate the electrodes’ photocurrent stability. The amperograms were obtained in transient condition for 300 s (5 s on/off—Figure 3b,c and Appendix A). As can be seen, all electrodes have a short response time (e.g., the time required for the photocurrent to reach its maximum value immediately after irradiation) (Table 1), and high stability and reproducibility (Figure 3b,c). Furthermore, it was possible to estimate the current density normalized by the photon flux (*j*_ph_). Its calculation can be performed by the difference between the current density during the radiation incidence with the absence. In this case, the estimated values were approximately 16, 8, and 0.7 µA cm^−2^ (amperogram in Figure 3c) for the electrodes ITO/FeVO(I),-(II),-(III), respectively (Table 1). This result highlights the photoelectrochemical properties of the iron vanadate electrodes, which can be associated with a greater surface homogeneity and high light absorption of the material under the ITO substrate. However, the amperogram in Figure 3c and the cyclic voltammograms (Figure 3a and Appendix A) clearly display a difference in photocurrent density between the electrodes. Even though the ITO/FeVO(II) starts with a higher *j*_N_ among the electrodes, there is a substantial decay during the measurement, and the electrode with five layers shows a higher photocurrent density.

Electrochemical impedance spectroscopy (EIS) was used to investigate the electrochemical characteristics of the electrode/solution interface and the relationship between the *j*_N_ with the number of deposition layers. From a mathematical adjustment to the simple Randles electrical circuit (inset Figure 3d), the Nyquist and Bode plots were useful to determine the charge transfer resistance (*R*_ct_) in the presence and absence of visible light. According to the CV profile in visible light and a previous manuscript [15], +0.45 V was selected for the measurements. The Nyquist and Bode plots of each electrode under the presence and absence of visible light, are provided in the Appendix A. Considering the obtained results under visible light, Figure 3d and Table 1, there is an increase in the semicircle size as the number of deposition layers increases, and consequently, the charge resistance is higher. The *R*_ct_ values follow the increase sequence: ITO/FeVO(III) (2160 × 10^3^ Ω) > ITO/FeVO(II) (257 × 10^3^ Ω) > ITO/FeVO(I) (119 × 10^3^ Ω). This result indicated that with a smaller thickness, the photoexcited electrons formed on the semiconductor surface could migrate more quickly to the substrate and, later, to the auxiliary electrode [54,55]. At the same time, in the dark condition, all *R*_ct_ values are extremely high (Table 1), due to the small formation of charge carriers. In Bode’s graphs, Appendix A, only a single peak was observed within the analyzed frequency range, which can be associated with charge transfer at the ITO/FeVO/electrolyte interface. A summary of the electrochemical impedance spectroscopy results for the measurements performed under the absence and incidence of light is shown in Appendix A.

All results confirmed the electrochemical parameters discussed in this manuscript. It was noticed how the visible light is a decisive feature for the ITO/FeVO electrodes properties, as well as the small thickness electrodes, such as the ITO/FeVO(I) electrode, showed a better photoelectrochemical response (higher photocurrent density, shorter response time, and lowest charge transfer resistance). This result may be associated with the highest incident light absorption, electrode thickness, and the homogeneity of iron vanadate semiconductor on the ITO substrate surface, as observed by SEM images. For these reasons, only the ITO/FeVO(I) electrode was investigated for the detection of dopamine.

### 2.3. Dopamine Detection

Subsequently, the electrocatalytic and photoelectrocatalytic activity of the ITO/FeVO(I) electrode was evaluated using the biomolecule dopamine (DOP). An aqueous solution of DOP was prepared in sodium sulfate (electrolytic solution) and used in different electrochemical procedures (cyclic voltammetry (CV), differential pulse voltammetry (DPV), and chronoamperometry). All measurements were performed in triplicate, and in the absence and presence of a visible light source (35 W Xe lamp–Vision HIDlamp).

Initially, a cyclic voltammogram was obtained (potential range between 0 and +1.0 V (vs. Ag/AgCl)) using the ITO/FeVO(I) electrode in 1.3 mmol L^−1^ DOP solution, and 0.1 mol L^−1^ Na_2_SO_4_ as a supporting electrolyte. As shown in Appendix A (solid red and black lines), two oxidation processes were observed around the potential of +0.3 and +0.6 V. More details are mentioned in the Appendix A. Furthermore, the current density (*j*_N_) was increased during the light incidence (solid red line), indicating that the visible light influences the oxidation process. The ITO photocurrent also was plotted in Appendix A to compare with the results. However, a negligible current was observed (<0.3 µA cm^−2^ at + 1.0 V (vs. Ag/AgCl)). Additionally, the obtained results were confirmed by another CV obtained at a lower DOP concentration level (0.15 mmol L^−1^—Appendix A).

However, cyclic voltammetry is frequently used in exploratory analyses, while differential pulse voltammetry (DPV) and chronoamperometry are the best options for electrochemical and photoelectrochemical detections (lower detection levels, higher sensibilities, and reproducible experiments) [53]. Thus, a DPV was firstly used to investigate the ITO/FeVO(I) electrode performance in DOP detection [13]. Sequential addition of the analyte was performed into the electrochemical cell (DOP concentrations ranging between 0 and 76 μmol L^−1^), and the respective voltammograms were recorded in the presence and absence of visible light (from 0.20 to +0.45 V), Figure 4a and Figure 4b, respectively. The DOP detection was accomplished by constructing analytical curves using the current densities at +0.30 V (Figure 4c).

As demonstrated in Figure 4, a linear increase in the current density (*j*_N_) response was observed in all voltammograms, particularly when the DOP concentration extended from 1.27 to 31.64 µmol L^−1^ (Figure 4a–c). Besides, a higher *j*_N_ was achieved during light incidence, essential to good sensibility and a low limit of detection (Figure 4a). The current densities were fitted to the linear model (Figure 4c), and the analytical parameters (linear range, slope or sensitivity, correlation coefficient, LOD, and LOQ) were estimated and are shown in Table 2. All obtained parameters were similar to other materials commonly used for dopamine detection [17,18,19].

Stability and interference studies were also performed. As shown in Figure 4d, a slight loss in the photoelectrochemical signal were obtained. The relative standard deviation (RSD) was calculated after seven successive essays, obtaining 2.5 and 3.4% for 10 and 20 µmol L^−1^ DOP solution, respectively (Figure 4d). These results revealed a good precision for the proposed method. Besides, the ITO/FeVO(I) electrode also had its photoelectroactivity evaluated in the presence of organic compounds that could be possible interferent in the commercial drug and human fluid sample analysis, such as ascorbic acid (AA), glucose (Glu), fructose (FR), uric acid (UA), and urea (UR). The tests were conducted with 10 µmol L^−1^ of DOP combined with the interferent organic compound (at the same concentration) (Appendix A). The signal obtained with bare DOP solution was compared with the signal from the mixture (DOP + interfering species), and it was possible to affirm that the response is only influenced by the DOP concentration in this condition (Appendix A).

Afterward, the chronoamperometry technique was additionally used for DOP detection. Chronoamperograms were obtained in transient light (chopped) condition. According to the cyclic voltammograms profile, 0.25, 0.30, 0.35, 0.40, 0.45, and +0.5 V (vs. Ag/AgCl) were choose for determining the more favorable potential in the dopamine oxidation reaction (Figure 5a,b). The tests were performed in the absence and presence of DOP solution (20 µmol L^−1^), using visible light for the electronic excitation (Appendix A). From the difference between the *j*_N_ with and without dopamine, +0.35 V was chosen as the best potential for dopamine detection, Figure 5a,b. From this potential, there is no significant gain in the current density that justify the use of more positive potential. Furthermore, this potential is in accordance with the onset potential determined by the CV curves (Figure 3a).

Analytical curves were built using the chronoamperometry technique at +0.35 V (vs. Ag/AgCl). The measurements were performed by DOP addition into the electrochemical cell under continuous stirring in the absence (Appendix A) and presence (Figure 5c) of visible light, to compare the electrochemical and photoelectrochemical activities (Figure 5d).

A fast and sensitive response was achieved to the successive addition of dopamine. Under visible light irradiation, the ITO/FeVO(I) electrode exhibited a wide linear concentration range (between 1.21 and 30.32 µmol L^−1^), with higher sensitivity (0.123 µA cm^−2^ µmol^−1^ L), and lower LOD value (0.34 µmol L^−1^) when compared to those obtained under dark condition (Table 3). Compared with the DPV method similarly studied in this manuscript, the chronoamperometry stands out with lower LOD and LOQ, and a brief higher sensibility (Table 2, Table 3 and Table 4). Besides, compared to other photoelectrochemical sensors for DOP detection, similar and, in some cases, better analytical parameters were obtained with the ITO/FeVO(I) electrode. For instance, in the work developed by Qin et al. [36], a microelectrode based on TiO_2_ nanotube and CdSe nanoparticles was constructed and evaluated in photoelectrochemical dopamine detection. The authors achieved a linear response in the 0.05 to 20 µmol L^−1^ dopamine concentration range, with 16.7 µmol L^−1^ as LOD. In another manuscript, dopamine detection was performed on anatase TiO_2_ nanoparticles sensitized with copper tetrasulfonated phthalocyanine (CuTsPc/TiO_2_) [56]. The photoelectrochemical measurements showed a linear response range between 4 and 810 µmol L^−1^ with a limit of detection of 0.5 µmol L^−1^. An additional remarkable amperometric sensor was obtained by Wang et al. [24]. The authors studied a FeOOH cocatalyst-modified nanoporous BiVO_4_ photoanode. Two linear ranges were observed (0.2–40 μmol L^−1^ and 40–1400 μmol L^−1^), and a low LOD was obtained (0.09 μmol L^−1^) for the photoelectrochemical dopamine oxidation. Furthermore, electrochemical sensors for dopamine detection were also compared with this work (Table 4).

The precision of the ITO/FeVO(I) electrode was assessed by detecting 10 and 20 µmol L^−1^ DOP solutions in intra- and inter-day repeatability studies (Appendix A). Considering intra-day precision (within-day), the RSD was calculated from ten successive assays, whereas the repeatability between days (inter-day) during five days of assays. As a result, the RSD of the current density obtained for intra-day precision was 2.10% (1.25 ± 0.06 µA cm^−2^) and 4.37% (2.39 ± 0.10 µA cm^−2^) for 10 and 20 µmol L^−1^ DOP solution, respectively (Appendix A). On the other hand, the RSD for inter-day precision was 1.80% (1.27 ± 0.02 µA cm^−2^) and 0.75% (2.39 ± 0.02 µA cm^−2^) for 10 and 20 µmol L^−1^ DOP solution, respectively (Appendix A). All RSD values indicated that the proposed chronoamperometric method presents good precision and stability. Furthermore, after five days, the current density response of the ITO/FeVO(I) electrode remained up at 99.1% and 98.7% of its initial value, considering 10 and 20 µmol L^−1^ DOP solution, respectively (Appendix A). All these measurements confirmed that the ITO/FeVO(I) electrode owns the desired repeatability.

Besides, the ITO/FeVO(I) electrode photoelectroactivity was additionally evaluated in the presence of organic compounds and body fluid. The interference study was conducted with 20 µmol L^−1^ of DOP combined with different organic compounds (glucose, fructose, uric acid, and urea) at the same concentration. As shown in Appendix A, the current is negligible at the initial stage (from 0 to 50 s). After light irradiation (from 50 to 100 s), there is a considerable gain in the current density due to the semiconductor excitation process. Afterward, with DOP addition (20 µmol L^−1^), the photocurrent reached a similar value as indicated in the analytical curve, even after the interfering species addition (GLU, FRU, UA, and UR, time from 100 to 350 s) (Figure 5d and Appendix A). In the final stage, around 350 s, dopamine was added again to the electrochemical cell. As a result, the photocurrent density was increased by the same level indicated on the analytical curve (Figure 5d and Appendix A). Thus, it is possible to affirm that the electrochemical response is only influenced by the DOP concentration in this condition.

Furthermore, the artificial cerebrospinal fluid (aCSF) was also used to evaluate the ITO/FeVO(I) electrode activity in an artificial biological system [57,58]. For more information about the aCSF preparation, please consult the Appendix A. Using 10 mL of a freshly prepared aCSF solution, 10 and 20 μmol L^−1^ of dopamine solution were added to the electrochemical cell, and the current density was monitored under the same experimental conditions. As can be seen in Appendix A, dopamine oxidation resulted in *j*_N_ very close to that obtained during the analytical curve construction. The recovery obtained was 102.4 and 98.7% for 10 and 20 μmol L^−1^ of dopamine solution, respectively, indicating no significant matrix interference effects and a good electrode performance, even in a more complex system (Appendix A).

The accuracy of the electrode and the proposed photoelectroanalytical method was evaluated by the determination of dopamine in a commercial drug sample. The sample analyzed is composed of levodopa + benserazide hydrochloride (Prolopa^®^ BD) and was chosen because of the structural similarities between levodopa and dopamine (please consult Appendix A). When metabolized by the human body, levodopa acts as a dopamine precursor [7,8,9,10]. A tablet containing around 100 mg of levodopa was evaluated, and the standard addition method was applied to compare with the medicine leaflet (Appendix A). As a result, the estimated levodopa concentration was around 18.9 μmol L^−1^, which represents 95 ± 2 mg of levodopa per tablet, with a relative error of 5% when compared with the label value. This result suggested that there is no significant difference between the levodopa and dopamine molecules signal, revealing a good performance of the ITO/FeVO(I) electrode in commercial drug analysis. This measurement was performed in similar photoelectrochemical conditions (visible light and applied potential of +0.35 V (vs. Ag/AgCl)).

DFT-based electronic structure calculations have been conducted to estimate the position of the frontier energy levels (HOMO and LUMO) of dopamine to help the interpretation of the oxidation processes. Appendix A illustrates the electronic levels of the working electrode (considering the FeVO_4_ + Fe_2_V_4_O_13_ mixture) and DOP species at distinct redox and protonation states (as well as expected degraded systems). The oxidation process can be rationalized by considering DOP_0_ and DOP_1_ structures, which are indeed the dominant species at the experimental conditions (pH ~5.4). The analysis of the local reactivity (CAFI) indicates that the oxidation of such species is mediated by effective interaction of the electrode and DOP resonant ring (please consult Appendix A, and the associated discussion). The improved light-induced DOP detection can also be interpreted in terms of charge transfer processes (Figure 6). In fact, an additional electronic transfer process can take place from the HOMO level of DOP_0_ due to light-induced transient changes of the electrode Fermi levels (splitting of the quasi-Fermi levels) and reduction of charge injection barriers (an issue that still deserves further investigation). Furthermore, the electronic structure calculations for organic species commonly present in human blood (α-glucose, β-glucose, D-fructose, L-fructose, urea, and uric acid—Appendix A), confirmed the interference study performed (Appendix A). In the experimental conditions, there is no favorable alignment for an electronic transfer process between the organic molecules and the ITO/FeVO(I) electrode (Appendix A).

## 3. Materials and Methods

### 3.1. Materials

Ammonium metavanadate (NH_4_VO_3_—Nuclear, 98.0%), ferric chloride hexahydrate (FeCl_3_.6H_2_O—Vetec, 98–102.0%), sodium sulfate (Na_2_SO_4_—Biotec, 99.0%), potassium chloride (KCl—Fmaia, 99.0%), potassium hexacyanoferrate(III) (K_3_[Fe(CN)_6_]—Fmaia, 99.0%), uric acid (C_5_H_4_N_4_O_3_—Sigma-Aldrich, 99.0%), L-ascorbic acid (C_6_H_8_O_6_—Synth, 99.0%), dopamine hydrochloride (C_8_H_11_NO_2_.HCl—Sigma-Aldrich, 99.0%), D-glucose (C_6_H_12_O_6_—Synth, 99.0%), and urea (NH_2_CONH_2_—Fmaia, 99.0–100.5%), were of analytical grade and used without prior purification. Besides, the Prolopa^®^ BD (levodopa + benserazide hydrochloride) (Roche) medicine was also used. The solutions were prepared using ultrapure water, with resistivity higher than 18.00 MΩ cm, obtained from a water purificator system (Elga model USF CE). The reagents were weighed using an analytical balance (Shimadzu AY 220). An oven (Brasdonto Model 5) and a muffle furnace (Edgcon 1P) were used for the heat treatment. The ITO glass conductive substrate (tin-doped indium oxide) used was purchased from Zhuhai Kaivo Optoelectronic Technology Co., Ltd. Company (Zhuhai, China) (<10 Ω sq^−1^ sheet resistance, transmittance >83%).

### 3.2. Construction of ITO/Iron Vanadate Electrode by the SILAR Process

The experimental procedure used to obtain the iron vanadate electrodes was the Successive Ionic Layer Adsorption and Reaction process (SILAR), based on the research group [15] and an adaptation by Tang and co-workers [41], with some modifications. Before the film deposition, the ITO conductive substrate (1.0 × 2.5 cm) was cleaned in an ultrasonic bath as follows: first, in deionized water for 30 min; second, in ethanol for 30 min; and finally, in acetone for 30 min. After the cleaning procedure, the substrate was dried in an oven at 100 °C for 30 min. In the sequence, two precursor solutions were prepared. The first solution (solution A) was prepared by dissolving 50 × 10^−3^ mol L^−1^ of NH_4_VO_3_ in 50 mL deionized water at 70 °C, resulting in a yellowish solution. Furthermore, the second solution (solution B) was prepared by dissolving 25 × 10^−3^ mol L^−1^ of FeCl_3_.6H_2_O in 50 mL deionized water at 25 °C, resulting in an orange color solution. Both solutions were maintained under magnetic stirring for 30 min. The deposition sequence was made as follows: immersion of the conductive substrate (1.0 cm^2^) in 10 mL of solution A, followed by immersion of the substrate in 10 mL of solution B, and ending with the washing in 10 mL of deionized water. For each solution, the electrode was maintained for 5 min. Therefore, one layer was obtained. This procedure was repeated five, ten, and fifteen times to obtain electrodes with different layers (for more details, please consult Appendix A). Finally, the electrode was thermally dried in air for 4 h and heated at 500 °C for 2 h in a muffle furnace in an air atmosphere. The temperature choice was due to the authors’ studies [41].

### 3.3. Physical Characterization

The physical characterization was carried out by X-ray diffraction (XRD), Fourier-transform infrared spectrophotometer (FTIR spectroscopy), diffuse reflectance, and scanning electron microscopy (SEM). Powder XRD data were obtained at room temperature on a PANalytical X′Pert PRO MPD diffractometer (Malvern Panalytical equipment–Almelo, The Netherlands) equipped with Cu-Kα radiation (1.54060 Å). The 2θ scan ranged between 5° and 80° with a scan step of 0.04° s^−1^ (6.0 s as counting time). To better understand some XRD parameters, the Rietveld refinement method was carried out (consult the Appendix A for more details). The infrared spectra were obtained in a Bruker Optik GmbH equipment (model Vertex 70/80, Ettlingen, Germany) ranging between 150 and 4000 cm^−1^. Reflectance accessory Platinum ATR was used, and the spectrums were acquired with 10 scans (resolution of 4 cm^−1^). The UV–Vis Shimadzu UV-2600 equipment (Kyoto, Japan) was used to estimate the band gap energy. The diffuse reflectance spectrum was obtained directly from the solid sample and the electrodes. Besides, the sample’s morphology was evaluated using the scanning electron microscope, model Quanta 200 produced by Philips/FEI Company (Hillsboro, OR, USA), applying a voltage of 25 kV, and the secondary electrons to generate the images.

### 3.4. Photoelectrochemical Characterization

The photoelectrochemical characterization was performed in a classical electrochemical quartz cell (20 mL), with only one compartment. An Ag/AgCl electrode (3.0 mol L^−1^ KCl), a platinum wire 10 cm, and a working electrode, the ITO/FeVO(I), -(II), -(III), were used as reference, auxiliary, and working electrode, respectively. A visible light source (35 W Xe lamp—Vision HIDlamp) was positioned 15 cm away from the electrochemical cell. Photoelectrochemical and electrochemical characterizations were conducted by cyclic voltammetry (CV) curves, chronoamperometry, and electrochemical impedance spectroscopy (EIS), using 10 mL of Na_2_SO_4_ 0.1 mol L^−1^, as the electrolytic solution. The procedures were controlled by the PGSTAT204 Autolab potentiostat/galvanostat (Barendrecht, The Netherlands), using free NOVA 2.1.5 software. The CVs were obtained in the potential range between 0 and +1.0 V (vs. Ag/AgCl), in the absence and light presence (20 mV s^−1^). Chronoamperometry curves and EIS measurements were obtained under different applied potentials (0, 0.2, 0.4, 0.6, 0.8, and 1.0 V (vs. Ag/AgCl)), under presence and absence of light. The Nyquist plots were obtained in the frequency range between 0.05 and 100 kHz, with an AC amplitude of 20 mV, and using Na_2_SO_4_ as an electrolytic solution. Furthermore, the electroactive surface area was also determined using the CV procedure and the Randles–Sevcik equation [53] (for more details, please consult the Appendix A).

### 3.5. Electrochemical and Photoelectrochemical Dopamine Detection

For electrochemical and photoelectrochemical dopamine determination, differential pulse voltammetry (DPV) and chronoamperometry techniques were used. DPVs voltammograms were obtained in the potential range between 0.20 and +0.45 V (vs. Ag/AgCl), with a modulation amplitude of 0.05 V, a step of 4 mV, and a modulation time of 0.05 s (these parameters were based on the Li et al. manuscript [13]). Chronoamperograms were obtained under different applied potentials (0.25, 0.30, 0.35, 0.40, 0.45, and 0.50 V (vs. Ag/AgCl)) for 150 or 600 s. All measurements were performed in the presence and absence of visible light, under different concentrations of dopamine solution.

The commercial drug sample of levodopa + benserazide hydrochloride (Prolopa^®^ BD) was used in the ITO/FeVO(I) electrode accuracy test. Firstly, a tablet (0.2742 g) was macerated, and a stock solution of approximately 1.33 mmol L^−1^ was prepared. For the standard addition method, 150 μL of levodopa diluted solution was added into the electrochemical cell, followed by successive addition of dopamine (5–25 μmol L^−1^). The photocurrent density was monitored at +0.35 V (vs. Ag/AgCl).

### 3.6. Molecular Modeling

Theoretical studies were conducted for dopamine at distinct redox, protonation, and states (DOP_0_ to DOP_4_) [59], including degraded structures (DOP_o-quinone_ and DOP_aminochrome_) [60] (see structures in Appendix A). To identify the most stable conformers, fifty distinct initial structures were considered for geometry optimization of unmodified DOP_0_, which were obtained via molecular dynamics (MD) calculations at high temperature, as described in ref. [61]. MD simulations were conducted with the aid of Gabedit computational package using AMBER force field [62]. The obtained structures were pre-optimized in a Hartree–Fock (HF) approach with semi-empirical PM7 Hamiltonian [63] and fully re-optimized in the framework of density functional theory (DFT) employing the B3LYP [64] exchange-correlation functional and 6-31G(d) basis set on all the atoms, considering water as a solvent (via PCM approach). Adsorption sites were identified via condensed to atoms Fukui indexes (CAFI) analysis, as described in the references [65,66,67]. HF/PM7 semiempirical calculations were conducted with the aid of MOPAC2016 software [68], and DFT/B3LYP/6-31G(d) calculations were conducted with the aid of the Gaussian 16 computational package [69].

## 4. Conclusions

The present manuscript shows a simple, efficient, and low-cost method of detecting dopamine using a photoelectrochemical platform. The SILAR process obtained the iron vanadate semiconductor material (FeVO_4_ and Fe_2_V_4_O_13_ phases) in the powder and film form (onto indium tin oxide (ITO) glass electrode surface). The material was critically characterized through X-ray diffraction (XRD), Fourier-transform infrared spectroscopy (FTIR), diffuse reflectance, and scanning electron microscopy (SEM) techniques. Under visible light, the photoanode with five layers (ITO/FeVO(I) electrode) showed the best photoelectrochemical response, such as higher photocurrent density, shorter response time, and lowest charge transfer resistance between the electrodes with 10 and 15 layers. Furthermore, the novel ITO/FeVO(I) sensor exhibited remarkable performance in dopamine detection. The reproducibility, selectivity, and accuracy were also evaluated, which indicated a good electrode performance with acceptable precision and no significant matrix interference effects. DFT-based electronic structure calculations were also employed to understand the electrode behavior in the presence of dopamine and other organic compounds. The improved light-induced DOP detection in terms of charge transfer processes was also discussed. Thus, from what was addressed, these results highlight the potential application of the assembled ITO/FeVO(I) platform as an alternative device for the dopamine photoelectrooxidation reaction.

## Figures and Tables

**Figure 1 molecules-27-06410-f001:**
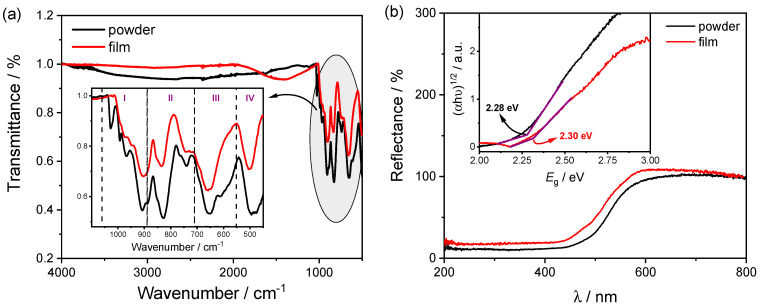
(**a**) Fourier-transform absorption infrared spectrum for the iron vanadate sample obtained by the successive ionic adsorption and reaction process. (**b**) UV–Vis diffuse reflectance and band gap energy evaluation from Wood–Tauc model. All measurements were performed for the powder and film iron vanadate.

**Figure 2 molecules-27-06410-f002:**
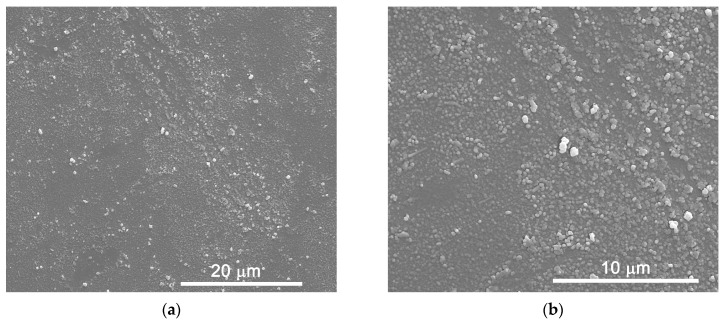
Scanning electron microscopy images for the iron vanadate sample obtained by the successive ionic adsorption and reaction process. The images were acquired using a magnification of (**a**) 5000×, (**b**) 12,000×, (**c**) 30,000×, and (**d**) 50,000×. The image (**d**) was obtained by the cross-section, with the average thickness of the electrode with five layers.

**Figure 3 molecules-27-06410-f003:**
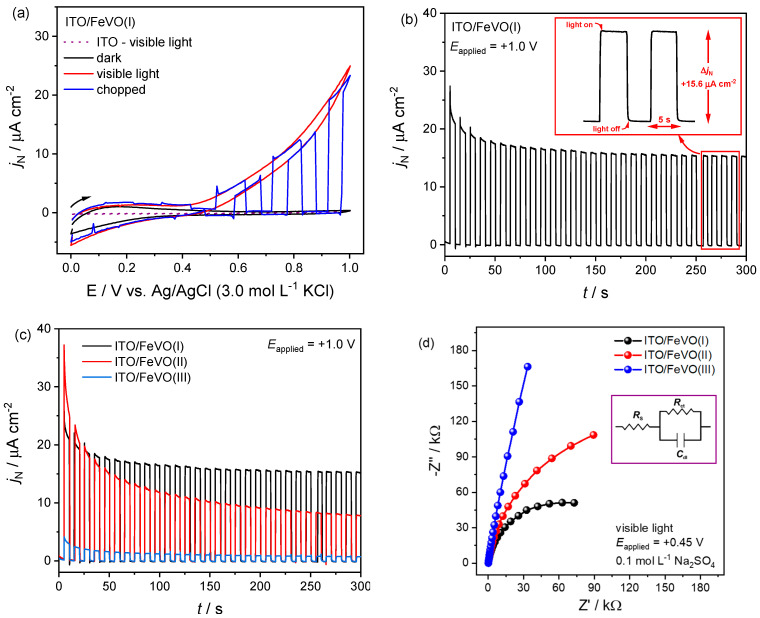
Photoelectrochemical measurements for the ITO/FeVO electrodes using cyclic voltammetry in the dark (solid black line), under continuous light illumination (solid red line), and under transient (chopped) light illumination (solid blue line) (10 s on/off) conditions. Scan rate of 10 mV s^−1^ (**a**). Photochronoamperogram under chopped light illumination (5 s on/off), at a +1.0 V (vs. Ag/AgCl), for ITO/FeVO(I) electrode (**b**), and for ITO/FeVO(I) -(II) and -(III) electrodes (**c**). Nyquist plot (electrochemical impedance spectroscopy), for the ITO/FeVO electrodes under visible light condition, at +0.45 V (vs. Ag/AgCl) (**d**). All measurements were performed in 0.1 mol L^−1^ Na_2_SO_4_ aqueous solution (pH = 5.4). Inset in (**b**), amperogram response under chopped light condition, and (**d**) simple Randles model circuit.

**Figure 4 molecules-27-06410-f004:**
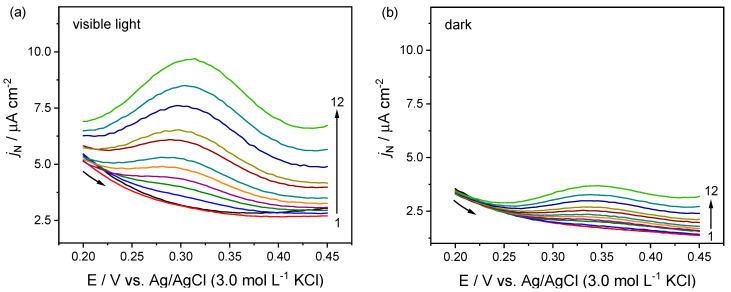
Differential pulse voltammograms obtained for dopamine in different concentrations (0, 1.27, 3.80, 7.59, 11.39, 15.19, 18.98, 25.31, 31.64, 44.29, 56.95, and 75.93 μmol L^−1^, points from 1 to 12) in (**a**) presence, and (**b**), absence of visible light (dark). Supporting electrolyte: 0.1 mol L^−1^ Na_2_SO_4_; linear range between 0.2 and +0.45 V (vs. Ag/AgCl). (**c**) Analytical curve obtained by linear adjustment. (**d**) Reproducibility test with 10 and 20 µmol L^−1^ dopamine concentration.

**Figure 5 molecules-27-06410-f005:**
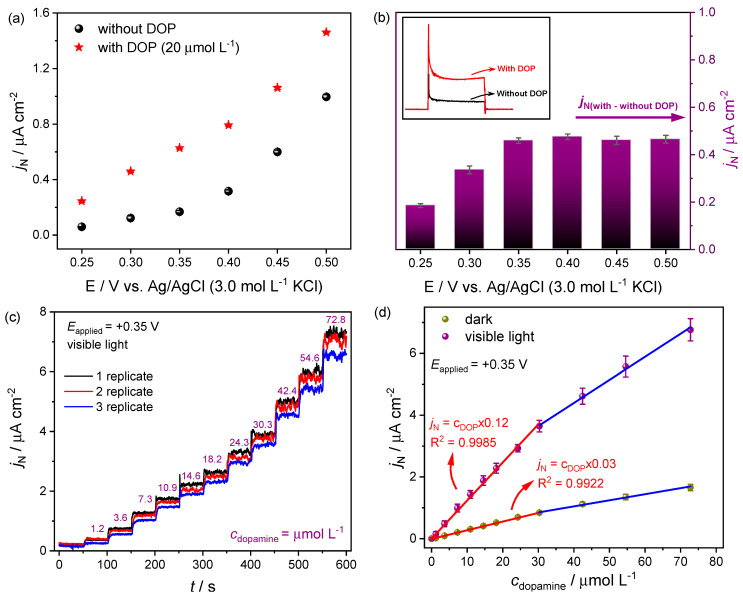
(**a**) Photocurrent density (*j*_N_) responses for chronoamperometry measurements under different applied potentials, and (**b**) the difference between the *j*_N_ with and without dopamine solution. Dopamine concentration: 20 μmol L^−1^. (**c**) Photocurrent density response obtained for different dopamine concentrations (from 1.21 to 72.77 μmol L^−1^). Measurement performed in triplicate and in the presence of visible light. (**d**) Analytical curve obtained by linear adjustment (calculation performed for the assays in the presence and absence of visible light). Supporting electrolyte: 0.1 mol L^−1^ Na_2_SO_4_.

**Figure 6 molecules-27-06410-f006:**
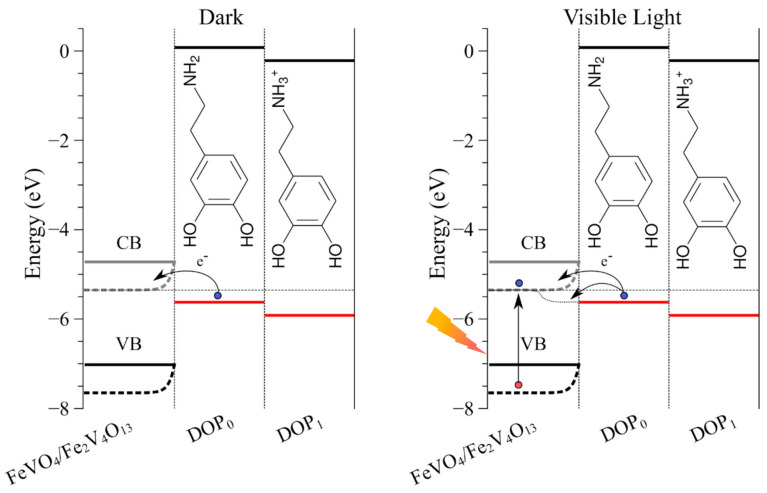
Relative alignment between the frontier electronic levels of electrodes and DOP species and illustration of charge transfer processes.

**Table 1 molecules-27-06410-t001:** Photoelectrochemical parameters of ITO/FeVO electrodes obtained by the SILAR process.

Electrode	EA/cm^2^	*j*_N_ (μA cm^−2^) ^[a]^		*j*_ph_ (μA cm^−2^) ^[b]^	Response Time ^[c]^/s	*R*_ct_ (kΩ) ^[d]^	
		Visible Light	Dark			Visible Light	Dark
ITO	0.44	0.30	0.20	~0.1	*	*	*
ITO/FeVO(I)	0.62	15.60	0.05	15.55	0.41	119	5525
ITO/FeVO(II)	0.50	7.80	0.10	7.70	0.51	257	2561
ITO/FeVO(III)	0.73	0.70	0.05	0.65	0.88	2160	2225

* Not determined. [a] At +1.0 V (vs. Ag/AgCl). [b] Current density obtained by the difference between *j*_N_ in continuous with dark condition at +1.0 V (vs. Ag/AgCl). [c] Average for response time of the electrodes at +1.0 V (vs. Ag/AgCl). [d] resistance of charge transference process.

**Table 2 molecules-27-06410-t002:** Analytical parameters for the voltammetric detection of dopamine using differential pulse voltammetric technique, and the ITO/FeVO(I) electrode obtained by the SILAR process (five layers).

Analytical Parameters	Condition	
	Dark	Visible Light
Linear range/µmol L^−1^	1.27–31.64; 31.64–75.93	1.27–31.64; 31.64–75.93
Sensitivity/µA cm^−2^ µmol^−1^ L	0.029	0.113
Correlation coefficient (R^2^) ^1^	0.9982	0.9987
Limit of detection/µmol L^−1^	0.76	1.71
Limit of quantification/µmol L^−1^	2.52	5.69

^1^ Data was fitted according to a linear model.

**Table 3 molecules-27-06410-t003:** Analytical parameters for the chronoamperometric detection of dopamine, using the ITO/FeVO(I) electrode obtained by the SILAR process (five layers).

Analytical Parameters	Condition	
	Dark	Visible Light
Linear range/µmol L^−1^	1.21–30.32; 30.32–72.77	1.21–30.32; 30.32–72.77
Sensitivity/µA cm^−2^ µmol^−1^ L	0.028	0.123
Correlation coefficient (R^2^) ^1^	0.9922	0.9985
Limit of detection/µmol L^−1^	0.33	0.34
Limit of quantification/µmol L^−1^	1.10	1.12

^1^ Data was fitted according to a linear model.

**Table 4 molecules-27-06410-t004:** Comparison of analytical conditions of ITO/FeVO(I) with other electrochemical and photoelectrochemical sensors for dopamine determination.

Sensor	Technique	Linear Range/µmol L^−1^	Limit of Detection/µmol L^−1^	Reference
CdSe/TiO_2_	PEC-CHRO	0.05–20.0	16.70	[36]
CuTsPc/TiO_2_	PEC-CHRO	4–810	0.50	[56]
FeOOH + BiVO_4_	PEC-CHRO	0.2–40; 40–1400	0.09	[24]
GO + Fe_3_O_4_	ELE-DPV	1–10	0.48	[17]
Fe_3_O_4_	ELE-SWV	5–50	7.10	[20]
Cu + GO	ELE-DPV	1–100	0.41	[21]
GO/WO_3_	ELE-CHRO	0.3–1245	0.31	[22]
ITO/FeVO(I)	PEC-DPV	1.27–31.64	1.71	This work
	PEC-CHRO	1.21–30.32	0.34	This work

PEC—photoelectrochemical; ELE—electrochemical; CHRO—chronoamperometry; DPV—differential pulse voltammetry; GO—graphene oxide; SWV—square wave voltammetry.

## Data Availability

The data presented in this study are available on request from the corresponding author.

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
