# Peer review of "Visible Light Photoelectrochemical Sensor for Dopamine: Determination Using Iron Vanadate Modified Electrode"

_molecules, 2022, doi:10.3390/molecules27196410_

Round 1
Reviewer 1 Report
The authors perform a nice study about FeVO4 as sensor for dopamin. They perform several electrochemical investigations for the characterisation of the material and some DFT calculations for the understanding. All in all, the study is solid and worth to publish. Some small comments:
- Fig. 2: Do not show the magnifications. Scale bars are enough.
- Show the comparison with other materials and studies in a Table with the performance. It makes easier for the reader to see the performance of the sensor with other materials.
- In my opinion, the conclusions are too enthusiastic. The electrodes show a good performance, but not an admirable performance.
- The authors should add the hints to supplementary material after the main text in the section about supplementary material.
- I do not understand why the data availability statement is not applicable. The authors should give the readers to see the raw data.
Author Response
Reply to the reviewers
Ref.: molecules-1920463
Title: "Visible Light Photoelectrochemical Sensor for Dopamine: Determination Using Iron Vanadate Modified Electrode"
Authors: Luan Pereira Camargo, Marcelo Rodrigues Da Silva Pelissari, Paulo Rogério Catarini da Silva, Augusto Batagin-Neto, Roberta Antigo Medeiros, Marcos Antônio Dias, Luiz Henrique Dall'Antonia
We are grateful to the reviewers for their comments that helped us to improve the paper. We believe that all the referees’ concerns were addressed below. We have modified the paper according to the reviewer’s comments. Modifications are highlighted in red in the new version of the paper.
Details are given as follows. The reviewer comments are reproduced below in italic, just before each of our answers.
Reviewers' comments:
Review Report Form (1)
Comments and Suggestions for Authors
The authors perform a nice study about FeVO4 as sensor for dopamin. They perform several electrochemical investigations for the characterisation of the material and some DFT calculations for the understanding. All in all, the study is solid and worth to publish. Some small comments:
- Fig. 2: Do not show the magnifications. Scale bars are enough.
Thanks for this observation. In this revised version, the correction was made.
- Show the comparison with other materials and studies in a Table with the performance. It makes easier for the reader to see the performance of the sensor with other materials.
We agree with the reviewer’s comment. In this revised version, we showed our responses and compared them with other materials in table form (please consult Table 4). All compared manuscripts are relevant and highly cited over time.
- In my opinion, the conclusions are too enthusiastic. The electrodes show a good performance, but not an admirable performance.
The observation made by the reviewer was very important. We have proposed a modification of the text, in agreement with the reviewer’s comment.
- The authors should add the hints to supplementary material after the main text in the section about supplementary material.
We would like to apologize for our mistake. In this revised version, we made the proper correction according to the Instruction for Authors (https://www.mdpi.com/authors/layout#_bookmark83).
- I do not understand why the data availability statement is not applicable. The authors should give the readers to see the raw data.
The data presented in this study will be available on request from the corresponding author. However, we are sending our raw data to the journal’s editor, and we will request to send the file to the reviewers (Data.xlsx).
We would like to thank the reviewer for all your comments. All the comments were very important to improve this new version of our article. We hope that this new version agrees with your notes.
Submission Date
31 August 2022
Date of this review
07 Sep 2022 17:23:10
Reviewer 2 Report
Journal Name: Molecules
Title: Visible Light Photoelectrochemical Sensor for Dopamine: Determination Using Iron Vanadate Modified Electrode
In the current research, the authors developed the photoelectrochemical sensor for detecting dopamine using the iron vanadate modified electrode system. This is very important research in the area of sensors and its scope matches the MDPI biosensors journal. But the article needs some major revisions to get published in MDPI biosensors.
I recommend its publication with major revision.
The authors electrodeposited the MoS2 nanoparticles on the Si surface for catalytical applications.
1. In the introduction the authors should write a comparison between the different methods for sensing dopamine using electrochemistry.
2. The auhthors need to read below mentioned articles and recommend to readers
A. https://pubs.acs.org/doi/abs/10.1021/acs.analchem.6b02481
B. https://www.sciencedirect.com/science/article/abs/pii/S0003267014012549
C. https://www.sciencedirect.com/science/article/abs/pii/S0167732217303057
D. https://www.sciencedirect.com/science/article/pii/S0925400517306664
E. https://www.sciencedirect.com/science/article/pii/S1388248114000228
3. I suggest authors cite relevant articles on MDPI Molecules so that it will prove the article’s suitability to publish on MDPI molecules.
4. What will be the practical applicability of the proposed method to sense dopamine in real samples like injection and blood serum since blood serum and dopamine injections are colored ones
5. The authors needs to perform frontier molecular orbital analysis/dual descriptor analysis or electrostatic potential maps and they need to compare their results with previously published articles on theoretical evidences.
6. The quality of the figures S9b is low
7. Graphical abstract is recommended
8. The authors need to compare the active surface area of bare ITO vs modified in a tabular format.
- Please check the grammatical and syntax error
Round 2
Reviewer 2 Report
The authors made sufficient changes article can be acceptable in the present form.